# Comprehensive Evaluation of Green Development in Dongliao River Basin from the Integration System of "Multi-Dimensions"

**Aoyang Wang** [1,2,3], **Zhijun Tong** [1,2,3,*], **Walian Du** [1,2,3], **Jiquan Zhang** [1,2,3], **Xingpeng Liu** [1,2,3] **and Zhiyi Yang** [4]

1   School of Environment, Northeast Normal University, Changchun 130024, China;
    wangay950@nenu.edu.cn (A.W.); duwl375@nenu.edu.cn (W.D.); zhangjq022@nenu.edu.cn (J.Z.);
    liuxp912@nenu.edu.cn (X.L.)
2   State Environmental Protection Key Laboratory of Wetland Ecology and Vegetation Restoration, Northeast
    Normal University, Changchun 130024, China
3   Laboratory for Vegetation Ecology, Ministry of Education, Changchun 130024, China
4   Changchun New Area Management Committee, Changchun 130024, China; xiaohaoyi2021@126.com
*   Correspondence: gis@nenu.edu.cn; Tel.: +86-1350-470-6797

**Abstract:** The bottlenecks in enhancing regional green development are resource shortages, environmental pollution, and ecological degradation. Taking the Dongliao River Basin (DRB) of Jilin Province as an example, this study explored green development from a multidimensional perspective. Based on the dimension evaluation results of REECC (resources, environment, and ecological carrying capacity), PLES (production–living–ecological space), and ER (ecological redline), the coupling coordination degree model and spatial autocorrelation model were constructed to explore the coupling coordination degree and spatial distribution of green development. The results showed that REECC had significant spatial differences, and the REECC index showed an increasing trend from northwest to southeast. In 2018, the overall level of green development in the DRB has obvious spatial dependence, but there were spatial differences, with a more obvious polarization from northwest to southeast. The spatial distribution of the coupling degree and coupling coordination degree is roughly the same, and there is a clustering distribution. The conclusions have practical significance for future environmental protection and economic production in the DRB.

**Keywords:** green development; spatial distribution; coupling coordination degree; Dongliao River Basin

## 1. Introduction

The natural environment is an eternal and necessary condition for the survival and development of human society and the natural basis for people's life and production. Development is the theme and common pursuit of human society [1]. Sustainable development can balance economic development and environmental protection while improving the level of social development [2]. With the progress of the social economy and the improvement of human spiritual civilization, sustainable development is constantly improving. Since the concept of sustainable development, with the goal of improving the quality of life of future generations, was put forward by the United Nations Conference on the Human Environment in 1972, countries have paid more attention to the sustainability of the earth. The sustainable development concept (SDC) is to achieve the goal of ecological environment and socioeconomic sustainability through the balance between humans and the ecosystem (dynamic balance) [3–5]. Green development (GD), as the second-generation theory of SDC, has gradually become the mainstream value orientation of global social economy and human development [6–8]. GD was first proposed in the last century and has experienced the enrichment of concepts such as green economy and ecological efficiency [9–13]. At present, it is generally believed that GD can be understood

as a means required to achieve a sustainable development in which human society realizes sustainable development under the constraints of ecological environment carrying capacity and resource carrying capacity on the premise of protecting ecological environment [14–16].

Due to the long-term imbalance between man and nature, an increasing amount of attention has been paid to GD and SDC, which have become important and meaningful topics in the 20th century. Scholars have begun to invest in research on GD. Internationally, research hotspots mainly include the green economy, supply chain, and urbanization [17,18]. China, as one of the developing countries, has many scholars that are concerned about GD, but the lack of efficient policies and practical actions has not led to the formation of national action sufficient to reverse the problem of ecology and environment. Although the economy is growing, the contradiction between China's environmental pollution, energy consumption, and economic development is becoming serious [19,20]. In order to solve these irreversible problems, the "China Human Development Report 2002", published by the United Nations Planning and Development Agency, pointed out that "green development" is the only way for China. To harmoniously develop the human environment in social and natural development, scholars have researched urban green development in the country or a larger area [21,22]. In the field of conceptual theory, scholars have discussed the conceptual and theoretical frameworks of GD [23,24]. In the practical application field, the research focus includes but is not limited to the construction of a GD evaluation system [25,26], measurement of the temporal and spatial relationship between GD level and economy, environmental governance, and policy intervention or industry [14,27,28].

The idea of ecological civilization was first put forward by President Xi at the 18th National Congress of the Communist Party of China. A unique opportunity is emerging in China, where all regions can comprehensively strengthen ecological environment protection and intensify pollution prevention and control, and the ecological environment has subsequently improved. However, the ecological environment is still not optimized. Ecological civilization is a new concept of economic development, and its specific application in China promotes sustainable development to the height of green development [29]. The Millennium Ecosystem Assessment (MEA) strived to link ecosystem services to human well-being [30], and The Economics of Ecosystems and Biodiversity (TEEB) added information about the dynamics of social ecosystems on this basis [31]. These studies are essentially to explore the path of sustainable development of the social–ecological system. At present, it is generally believed that comprehensive planning and policy tools to coordinate ecological, environmental, and socioeconomic needs are the most effective ways to solve the dilemma of protection and development [32]. The emergence of green trends has promoted the emergence of the ecological redline paradigm in natural resources and ecosystem management [33]. Although there is no definite criterion to connect the science research to policy making, China's ecological redline policy can provide a policy basis for effectively solving the problem of sustainable development [34,35]. On the basis of GD, China also put forward the concept of production–living–ecological spaces (PLES) policy, that is, to moderately centralize production space, promote efficient and intensive production space and organic concentration of living space, slow down the squeeze of production on living space and ecological space and return the saved production and living space to farmland and forests, and expand ecological space [36]. A major challenge is that various problems such as the imbalance of regional resource supply and demand, environmental pollution, and ecological degradation have gradually intensified, which is difficult to balance because the social economy and natural environment are constantly developing and changing [14]. Therefore, it is imperative to construct ecological civilization to solve problems related to natural environment and resources between regions, and a comprehensive evaluation of the regional GD status is required. In 2016, China's National Development and Reform Commission announced the "Green Development Index System", which fully reflects the connotation of the new concept of green development Quality and ecological civilization construction requirements. Resource, environment, ecology, spatial utilization of production–living–ecological spaces, and ecological redlines

are indicators that can best represent the degree of GD in ecosystems where humans and nature coexist on the basis of the "Green Development Index System". The coupling system of the three and their coupling coordination degrees can show the regional GD status in a multidimensional space.

As one of the largest commodity grain production bases in China, the Dongliao River Basin (DRB) is a regional political and economic center with farming as its main economic value, which faces great challenges between social–economic development and ecological protection. The Dongliao River Basin has problems such as water pollution, uncoordinated land use, and a relatively fragile ecosystem [37,38]. The study of regional GD can not only provide reference and enlightenment for regional policy design along the Dongliao Agricultural Economic Belt but, also, a scientific basis for future policy planning for the basin. This research can provide a scientific basis for the coordinated development of the river basin economy and the ecological environment which is based on agricultural production. This paper mainly explores the degree of GD in the DRB in the context of resource shortages, environmental pollution, and ecological degradation. Coupling evaluation of the GD degree has a certain reference significance for realizing sustainable development in this region. Therefore, it is of great significance to conduct in-depth evaluation and research on GD to effectively promote the green transformation of each administrative division and provide a reference and demonstrations for other river basins to explore GD paths.

In recent years, spatial autocorrelation analysis has proven to be one of the most important methods for exploring the spatial correlation of adjacent locations [39,40]. The Moran index, as an important method of spatial autocorrelation, has been widely used to characterize the degree of land use and vegetation coverage because of its relatively fast and automated advantages [41–43]. However, few studies have focused on spatial changes in GD using the spatial autocorrelation model, although it is effective and important to study the comprehensive situation of GD by using spatial autocorrelation, which can quickly obtain information about spatial clustering and, thus, evaluate the system and stability. GD is a system composed of the natural environment, society, and economy, but few studies have analyzed its evolution mechanism from multiple dimensions [44]. In summary, although recent research on comprehensive carrying capacity and green capacity has achieved great results, there is a lack of further expansion of the perspective of regional GD, and few studies consider resources, environment, ecology, socioeconomics, land and resources planning, and policies as coupling evaluation perspectives [45–47]. This paper combines three evaluation concepts and adopts the "multi-dimensions" integration technology, which improves the subjective and single problems in the previous evaluation of green development in a certain area.

Several important contributions to green development research were made in this study: (1) Green development was evaluated from a new perspective and its definition and connotations enriched. (2) An evaluation model was established for the coupling and coordination degree of regional watershed green development covering the four aspects of resources, environment, ecology, and social economy, and was made into a measure of the consistency and benign interactions between the various subsystems (REECC, PLES, and ER subsystems) of the green development system. (3) Moran's index was applied to explore the spatial autocorrelation of GD. Based on the evaluation results, we provide policy suggestions for GD in the DRB.

Therefore, the objectives of this study were (1) to construct an evaluation index system of comprehensive carrying capacity in the Dongliao River Basin by coupling resources, environment, ecology, and socioeconomics; (2) to analyze the spatial distribution of comprehensive carrying capacity, production–living–ecological space and ecological redline; (3) to reveal the coupling and coordination relationship between comprehensive carrying capacity, production–living–ecological space and ecological redline; and (4) to discuss the current situation of green development under the "multi-dimensions" integration system and provide policy suggestions.

## 2. Materials and Methods

### 2.1. Study Area

The Dongliao River originates from Liaoheyuan Town, Dongliao County, Jilin Province, China, and flows through Liaoyuan, Dongfeng, Dongliao, Yitong, Lishu, Gongzhuling, Shuangliao and Siping, Jilin Province (Figure 1). The Dongliao River Basin covers an area of 11,306 km$^2$, which is mainly cultivated land and forest land. The Dongliao River Basin is an important commodity grain production base in China, mainly planting corn, soybeans, rice, etc. It is also a highly densely populated area in Jilin Province. In the period of rapid socioeconomic development in the 20th century, the area has experienced serious water pollution, reduced resource and environmental capacity, shrinkage of wetlands and forests, soil erosion, unreasonable resource allocation, and backward ecological environment management models which, in turn, have restricted the economic growth of the region due to natural and human irrational activities. As a key river basin in Jilin Province and even the whole country, the Dongliao River Basin has been regarded as a demonstration area for ecological restoration since 2007. At present, the pollution situation in the area has been preliminarily controlled and scientifically treated.

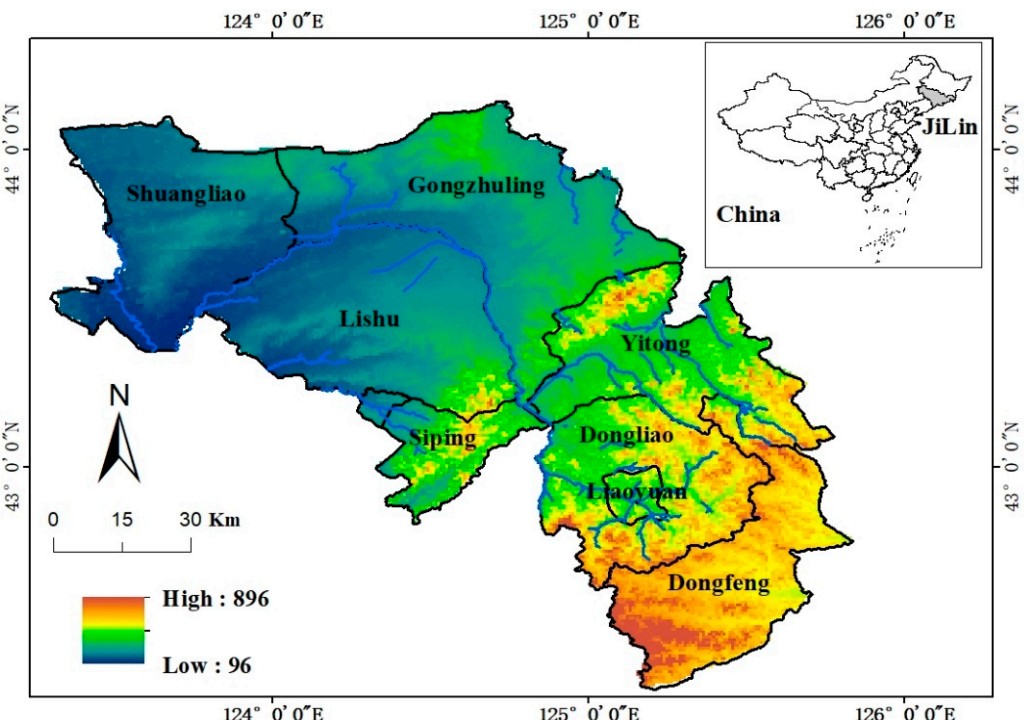

**Figure 1.** The location of the Dongliao River Basin.

### 2.2. Data Sources

The land use data were obtained from the Resource and Environmental Science and Data Center (RESDS, http://www.resdc.cn, accessed on 26 March 2021). The ecological environment data were obtained from the Jilin Province Water Resources Bulletin and Jilin Province Environmental Quality Report. Socioeconomic data are from the Siping Statistical Yearbook, Liaoyuan Statistical Yearbook, Jilin Statistical Yearbook, and Statistical Bulletin of National Economic and Social Development. In addition, some index data were calculated by the author according to the data. Furthermore, the ecological redline data required in the research is delineated by the author according to the "Three Lines and One List" compilation guide.

### 2.3. Methodology Framework

According to related research, regional GD is defined as a mode in which the regional resources–environment–ecology–socioeconomic system can guarantee local development

in a certain period under the constraints of rational resource development, environmental protection, ecological restoration, and socioeconomic development. This study applied the coupling model of REECC–PLES–ER to the spatial quantitative study of the green coordinated development of the DRB in 2018, the dynamic evolution of the coupling system was simulated, the important influencing factors and state factors were identified, and scientific suggestions were put forward for region GD. The methodology framework is illustrated in Figure 2.

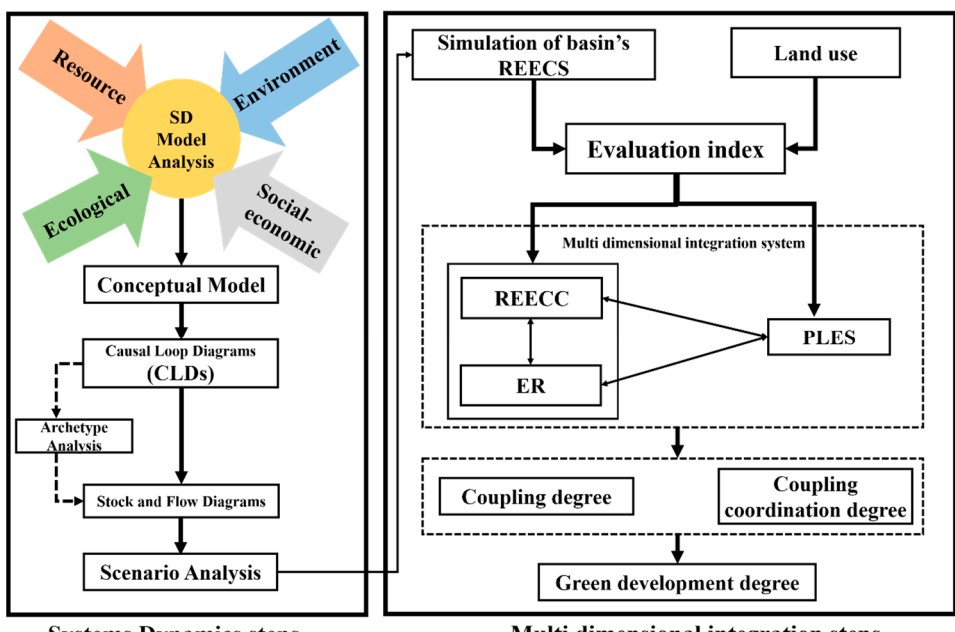

**Figure 2.** Research framework.

2.3.1. System Dynamics Model (SD Model)

System dynamics (SD) is a discipline that analyzes and simulates dynamic complex systems. It is an extremely effective tool and simulation method for understanding and dealing with high-order, nonlinear, and multi-feedback time-varying systems. The research objects of SD are mainly open, nonlinear, high-level, multivariable, multi-feedback, complex, and time-varying systems. The research purpose of SD is to select the leading factors affecting the development of the system by analyzing the causal feedback relationship in the system and the influence of many factors on the system objectives and, finally, to provide a scientific decision-making basis for decision-makers. Therefore, the SD model is highly applicable to an integrated system of resources, environment, ecology, and social economy, which pursues the best goal of the whole system dynamics and emphasizes the coordinated development of each subsystem.

According to the system dynamics model, the resource dimension refers to the quantity and quality of regional resources, the environmental dimension is the space for human activities, and the ecological dimension is a comprehensive system that organically combines the natural environment and social economic activities. The resource dimension is the basic condition of the ecological dimension, and the environmental dimension is the constraint condition of the ecological dimension. According to the analysis of the characteristics of resources, environment, and ecosystem in the basin and the complex relationships among the factors in the system, Figure 3 was drawn. This study established a dynamic simulation model of resources, environment, and ecosystem in the region according to the principle of system dynamics and established a causality diagram and structural flow diagram of the SD model in the basin by using VENSIM-DSS.

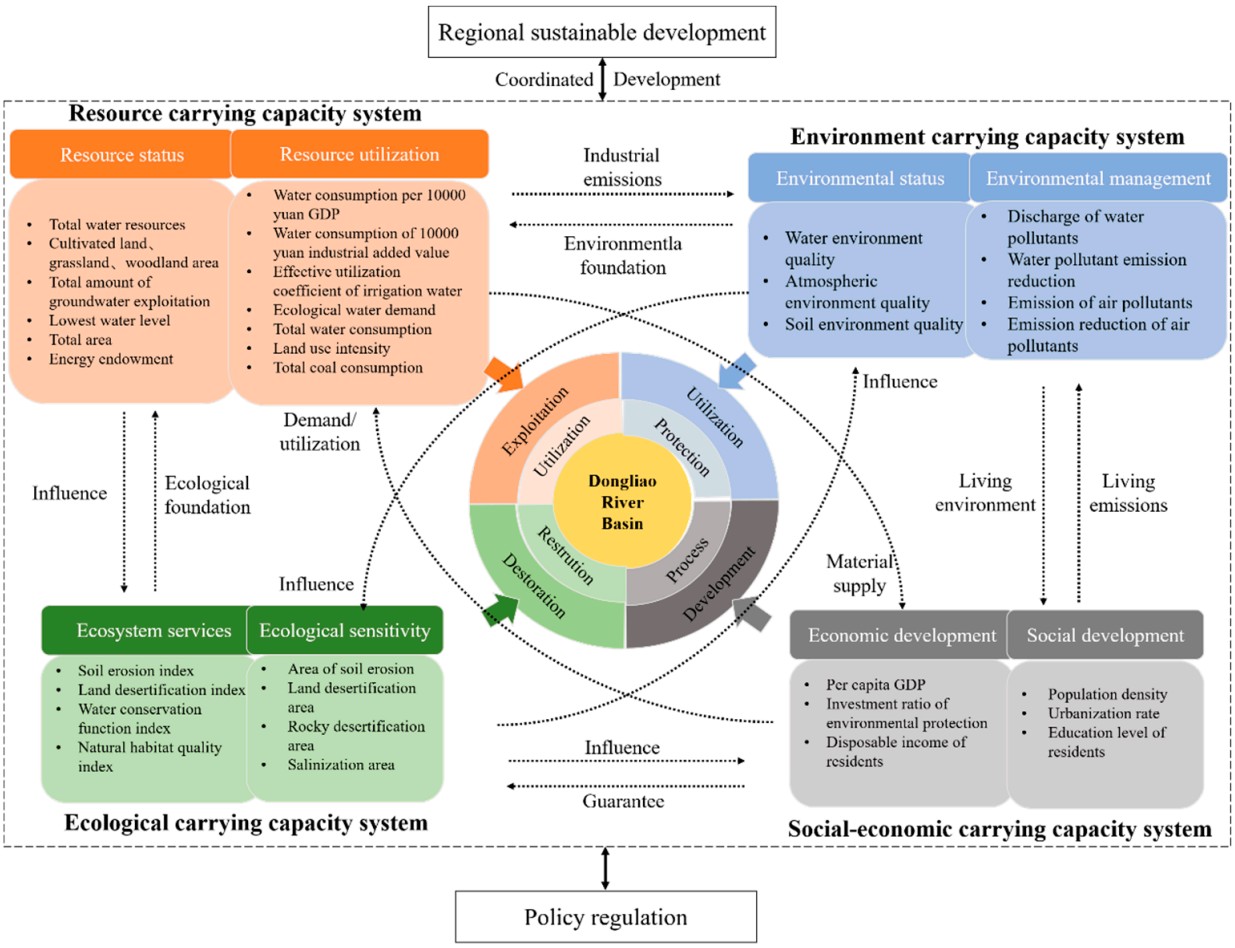

**Figure 3.** Regional REECC structural diagram.

### 2.3.2. Weight Determination

Considering the complexity and uncertainty of the resource–environment–ecology–socioeconomic system, a combination of principal component analysis and geodetector were used to improve the objectivity of carrying capacity evaluation. Principal component analysis, with the advantages of subjectivity, can determine the weight of indicators by calculating the information entropy, and the indicators with greater variation have greater weights. Factor detection in a geodetector can analyze the degree of interpretation of each factor to state variables. In this study, the weight distribution of state indicators (S) was determined by principal component analysis, and the interpretation degree of pressure (P) and response (R) indicators to state variables was then obtained by using the geodetector model, which was used as the weight. This method is relatively objective, can be widely used in various fields, and has strong research value.

### 2.3.3. Delineation of Production–Living–Ecological Space

This paper refers to the national standards of land function status classification and related research and combines expert opinions to establish a scoring standard system for the PLES in the Dongliao River Basin based on land use data. This system divides farmland into production space, construction land into living space, and other land use types into ecological space.

### 2.3.4. Construction of Coupling Coordination Degree Model

From the perspective of "multi-dimensions", the coupling coordination degree of regional green development consists of three parts: resource, environment, and ecological

carrying capacity subsystem, production-living–ecological space subsystem, and ecological redline subsystem. Based on the coupling relationship of mutual influence and mutual restriction in each space, this study introduces the coupling coordination degree model used in physics. The study uses the coupling degree to express the interaction level of the system, taking advantage of the coupling coordination degree to express the comprehensive coordinated development among the three to discuss the spatial distribution and regular characteristics of the coupling coordination relationship among *REECC*, *PLES*, and *ER*. The coupling degree was calculated using the following formula:

$$C = \left\{ \frac{REECC \times PLES \times ER}{[REECC + PLES + ER]^3} \right\}^{1/3} \tag{1}$$

*C* refers to the coupling degree of *REECC–PLES–ER* system in each unit, $C \in [0, 1]$. The value of *C* reflects the degree of correlation between systems. On the basis of formula (1), a coupling degree model is given to reflect the interaction intensity of *REECC–PLES* function (*C1*), *REECC–ER* function (*C2*) and *PLES–ER* function (*C3*):

$$C_1 = 2 \times \left\{ \frac{REECC \times PLES}{[REECC + PLES]^2} \right\}^{1/2}$$

$$C_2 = 2 \times \left\{ \frac{REECC \times ER}{[REECC + ER]^2} \right\}^{1/2} \tag{2}$$

$$C_3 = 2 \times \left\{ \frac{PLES \times ER}{[PLES + ER]^2} \right\}^{1/2}$$

The following coupling coordination model was adopted: $\alpha$, $\beta$, and $\gamma$ are the corresponding weights of each subsystem. The functional indices of REECC, PLES, and ER are equally important in this paper, so $\alpha = \beta = \gamma = 1/2$.

$$T = \alpha REECC + \beta PLES + \gamma ER$$

$$T_1 = \alpha REECC + \beta PLES$$

$$T_2 = \alpha REECC + \gamma ER \tag{3}$$

$$T_3 = \beta PLES + \gamma ER$$

$$D_i = \sqrt{C_i \times T_i}$$

where *T* is the comprehensive evaluation index of GD, $T \in [0, 1]$, and the coupling coordination degree *D* between systems is classified on the premise of calculating the coupling coordination degree. By referring to the existing research results and combining the research results, the coupling coordination degree can be divided into five types: extremely low, low, middle, high, and extremely high.

2.3.5. Moran's Index

Geographical spatial autocorrelation refers to the correlation between adjacent values in a time series, and its measure can be divided into global spatial autocorrelation and local spatial autocorrelation [48,49]. The local Moran index can calculate the degree of aggregation and identify the specific spatial location of anomalies. Therefore, this study uses GeoDa1.16, selects Moran's I index to represent the aggregation of explained variables as a whole, and the local spatial correlation index (LISA) tests the spatial autocorrelation of explained variables, which is used to measure the aggregation or dispersion effect of

low and high values in space to measure the correlation degree of spatial functions in the region [50–52]. The calculation formula is as follows:

$$Global\ Moran's\ I = \frac{\sum_{i=1}^{n}\sum_{i=1}^{n}(x_i - \overline{x})(x_j - \overline{x})}{s^2 \sum_{i=1}^{n}\sum_{i=1}^{n}\omega_{ij}} \tag{4}$$

Among them, n is the number of regional units, $x_i$ and $x_j$ are the observed values of regions *I* and *j*, $s^2$ is the variance of samples, and $\omega_{ij}$ is the spatial weight matrix. The values of *Global Moran's I* are distributed in [1, 1], and when its value is greater than 0, it indicates a positive spatial correlation, and the values of variables in adjacent spatial points have high similarity and aggregation occurs between high values. When its value is less than 0, it indicates there is aggregation between high and low values. The larger the absolute value, the higher the degree of spatial autocorrelation.

$$Anselin\ Global\ Moran's\ I = \frac{(x_i - \overline{x})}{s^2 \sum_{i=1,j\neq 1}^{n}\omega_{ij}(x_i - \overline{x})} \tag{5}$$

In the formula, *n* is the number of regional units, $x_i$ is the observed value of region *I*, $s^2$ is the variance of the samples, and $\omega_{ij}$ is the spatial weight matrix. The value range of the local Moran index is not limited to [–1, 1]. The high absolute value of the local Moran index indicates that the area units with similar variable values are clustered in space, and the low absolute value indicates that the area units with dissimilar variables are clustered in space.

### 3. Results

#### 3.1. Construction of the SD Model for Regional REECC

REECC is a comprehensive system, and the relationships among its subsystems are complex. First, the boundary of the total system should be determined, and then the interaction relationship among subsystems, the main factors, and the causal feedback relationship within each subsystem should be determined, which will lay the foundation for the next design of the model. According to the positioning and requirements of sustainable development in this region, the entire system is decomposed into four subsystems: resource carrying capacity, environmental carrying capacity, ecological carrying capacity, and socioeconomic carrying capacity. The subsystems interact with each other and cause and affect each other, forming a loop diagram with multiple feedbacks, as shown in Figure 4.

#### 3.2. Construction of A REECC Evaluation Index System

The resource–environment ecosystem is not only a comprehensive system composed of economic, social, environmental, ecological, and resource subsystems but also a dynamically evolving composite system. The evaluation index system of REECC should not only comprehensively reflect the status and characteristics of each subsystem, but also reflect the action mechanism of resources and environmental ecosystems. Therefore, this study divides the whole system into resource, environment, ecological, and socioeconomic subsystems based on the coupled model framework and describes the characteristics of the system and the interaction between the systems from four dimensions. Taking pressure (P) and response (R) as input factors is the index of influencing factors of resources, environment, and ecosystem. State (S) is an observable and directly apparent factor in the ecological security system, which is directly and indirectly affected by output factors. In this study, the state (S) is taken as the result measurement index, which shows the interaction among the system elements and comprehensively reflects the overall situation of the bearing capacity system. Taking into account the large differences in the selection of some indicators in the SD conceptual model and the lack of some statistical indicator data, combined with the environmental damage and ecological degradation of the DRB and the statistical data of Chinese cities. On the basis of the "Green Development Index System" announced by China's National Development and Reform Commission, the indicator level in this paper was selected based on the principles of difference and feasibility according to

scientific, comprehensive, and the evaluation index, which is constructed from three levels: objection level, criterion level. The evaluation index system in DRB is composed of four result measurement indexes and 11 influencing factor indexes (as shown in Table 1). The weight of the state layer in the indicator was obtained by principal component analysis, and the results are shown in Table 2.

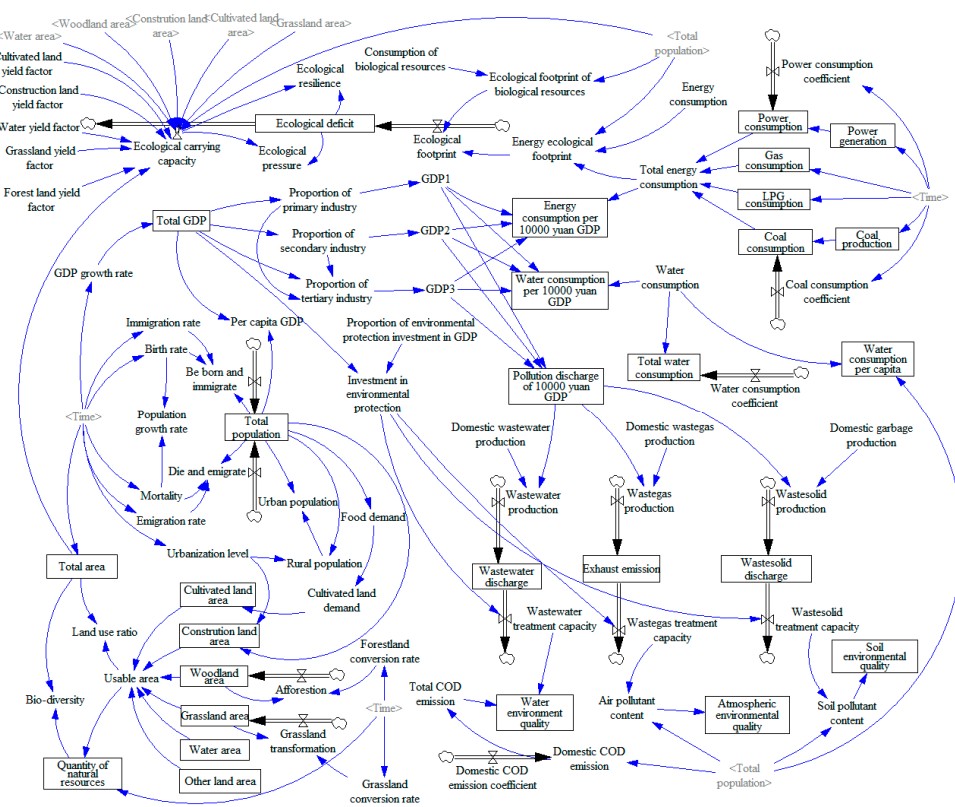

**Figure 4.** Causal feedback flow chart of GD system of DBS.

**Table 1.** Evaluation index system of REECS.

| Object Hierarchy | Criteria Layer | Indicators | Serial Number | Weight | Attributes |
|---|---|---|---|---|---|
| REECC in DBS | State B1 | Ecological environment quality index | C1 | 0.2249 | + |
| | | Air pollution index | C2 | 0.4035 | - |
| | | Water quality index | C3 | 0.2470 | - |
| | | Total ecological redline index | C4 | 0.1246 | - |
| | Pressure B2 | Salinization | C5 | 0.0809 | - |
| | | Desertification of land | C6 | 0.0911 | - |
| | | Soil erosion | C7 | 0.0886 | - |
| | | Wind prevention and sand fixation | C8 | 0.0867 | - |
| | | Conservation of water and soil | C9 | 0.1218 | - |
| | | Water conservation | C10 | 0.0909 | - |
| | Response B3 | Sewage discharge | C11 | 0.0819 | - |
| | | Sewage treatment capacity | C12 | 0.0704 | + |
| | | Exhaust emissions | C13 | 0.0600 | - |
| | | Nature reserve | C14 | 0.1492 | - |

**Table 2.** Component matrix of the state layer.

| Indicators in State Layer | Principal Component | |
|---|---|---|
| | PC1 | PC2 |
| Ecological environment quality index | 0.9248 | −0.2444 |
| Air pollution index | 0.9060 | −0.2514 |
| Water quality index | 0.5546 | 0.3827 |
| Total ecological redline index | 0.2797 | 0.8634 |

### 3.3. Spatial Distribution of REECC, PLES, and ER

The results of spatial distribution completed with Arcgis 10.7 are shown in Figure 5, where the REECC of the study area presents spatial distribution characteristics of decreasing gradually from southeast to northwest, with significant regional differences. The extremely high, high, medium, low, and extremely low-level areas of REECC present a mass distribution of "large clusters, small dispersions, and patches" in space. Relying on the drive of the central city and the support of characteristic industries and township enterprises, the area with extremely high or high levels of REECC present scattered distribution characteristics, mainly in the Dongfeng–Dongliao area in the southeast. Areas with low carrying capacity are either limited by terrain, inconvenient transportation, inconvenient regional superiority, or lack of advantageous resources to support development, pillar enterprises, weak population agglomeration capacity, poor economic foundation, and regional distribution characteristics located in the Shuangliao–Gongzhuling area.

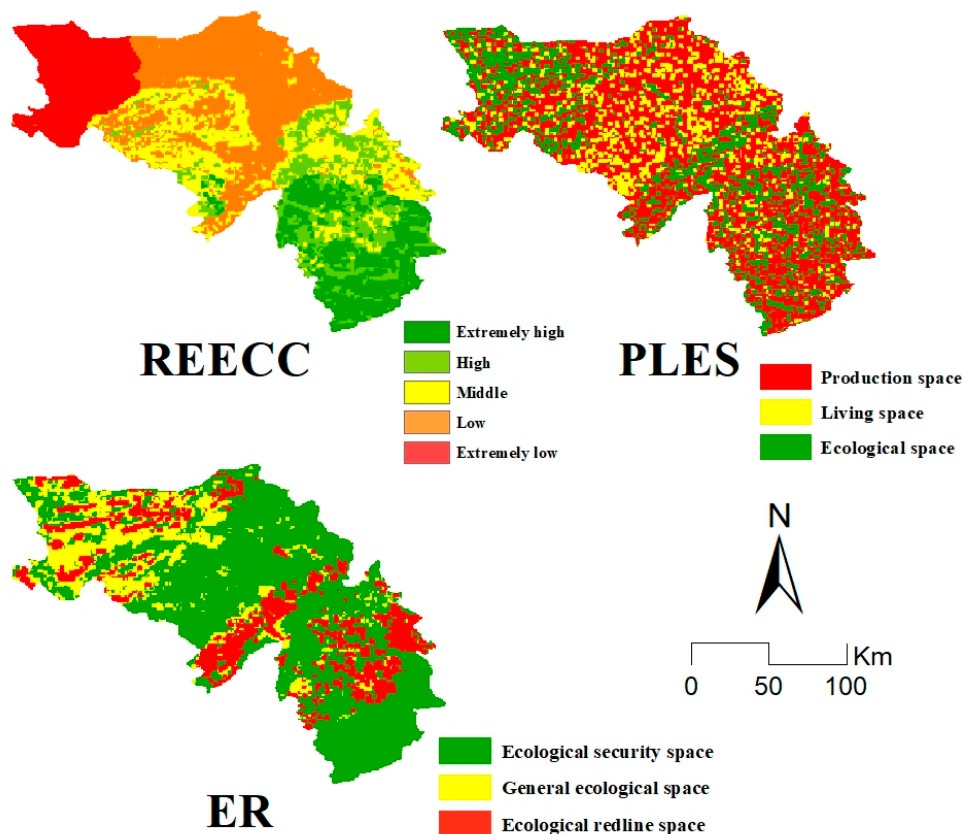

**Figure 5.** The spatial distribution of regional REECC, PLES, and ER.

Since land use in the DRB is mainly cultivated land, the overall spatial pattern of PLES presents the characteristics of "large production space, small point living space, and small scale and low value flaky ecological space". Among them, "large production space" mainly

refers to most towns in Gongzhuling City, Lishu County, Dongliao County, and Dongfeng County in Siping and Liaoyuan. This region is the core area of regional crop economic development in the DRB. "Small point living space" refers to the human living space scattered in the whole river basin. "Small scale low value flaky ecological space" is located in the west of Shuangliao City and Yitong County, accounting for a small part of the whole watershed, which shows the unbalanced spatial pattern of production–living–ecological spaces at the county level in the DRB. The spatial distribution of the ecological redline shows that it is mainly located in the east of Siping City and Lishu County, and most of Liaoyuan City and Yitong County. In addition, Shuangliao City and Gongzhuling also have sporadic distributions. The general ecological zone is mainly located in most of Shuangliao City and the west of Gongzhuling City. Most of the remaining areas are ecological security areas, which shows that the ecological security of the DRB is good.

*3.4. Comprehensive Index of Green Development of Resources, Environment, and Ecosystem in DRB*

The comprehensive index of green development was high in 2018, indicating that the overall pressure on the ecological environment in this period was relatively small and the degree of green development was relatively high (Figure 6). The results of spatial differentiation show that there are significant differences between urban and rural areas; during the study period, almost all cities were at the middle and lower levels, while almost all rural areas were at the middle and above levels. Among them, the areas with the highest GD index are located in Dongliao County and Dongfeng County, while the areas with the lowest GD index are Shuangliao City, Siping, and Liaoyuan.

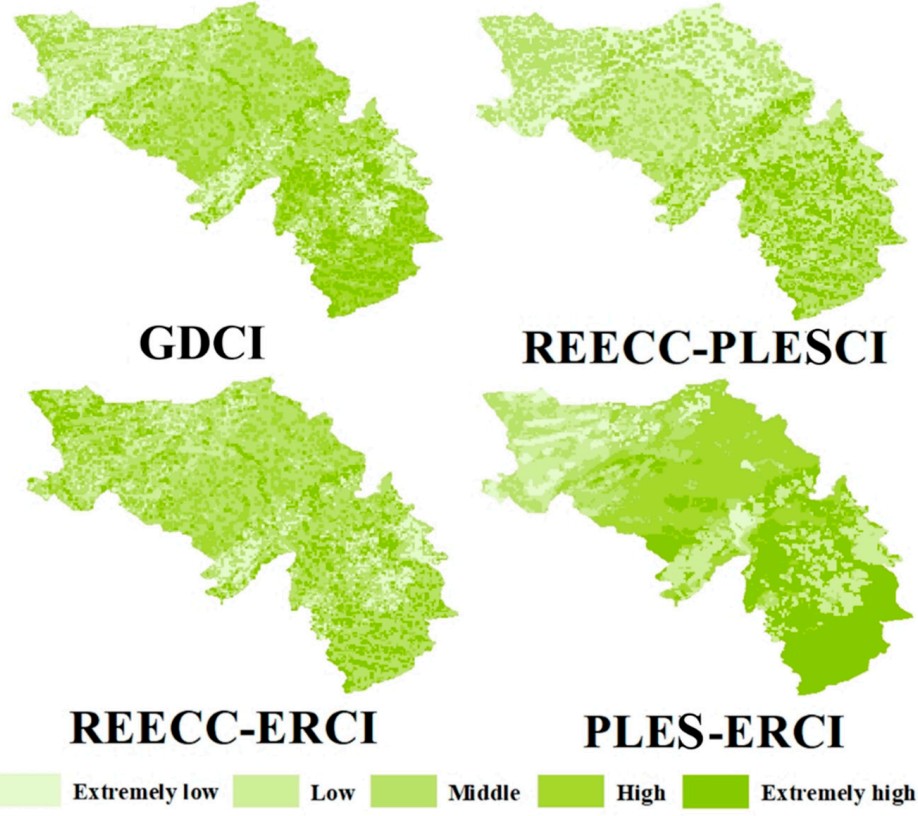

**Figure 6.** The spatial distribution of green development composite index.

## 4. Discussion

To show the spatial characteristics of the comprehensive index of green development of resources, environment, and ecosystem more intuitively, a spatial distribution map of LISA was drawn using Geoda (Figure 7). The global Moran index is positive, but HH

clusters are mainly distributed in the central areas of Liaoyuan and Siping, while LL clusters are distributed in Shuangliao City, west of Gongzhuling City, and east of Siping City and Lishu County, which indicates that there is an obvious agglomeration phenomenon in areas with similar comprehensive indices of GD in the study area. According to the calculation of the comprehensive index of green development of the coupled system, the spatial state of resources, environment, and ecosystem is consistent with the ecological redline and production–living–ecological spaces. This reflects, to a certain extent, that land use and ecological function are the leading factors affecting the comprehensive index of GD in the DRB; that is, the increase or decrease of green development degree is mainly determined by local land use and ecological protection. Therefore, in 2018, the spatial function utilization of production–living–ecological spaces was relatively coordinated, the ecological protection situation was generally average, and the green development situation was more optimistic.

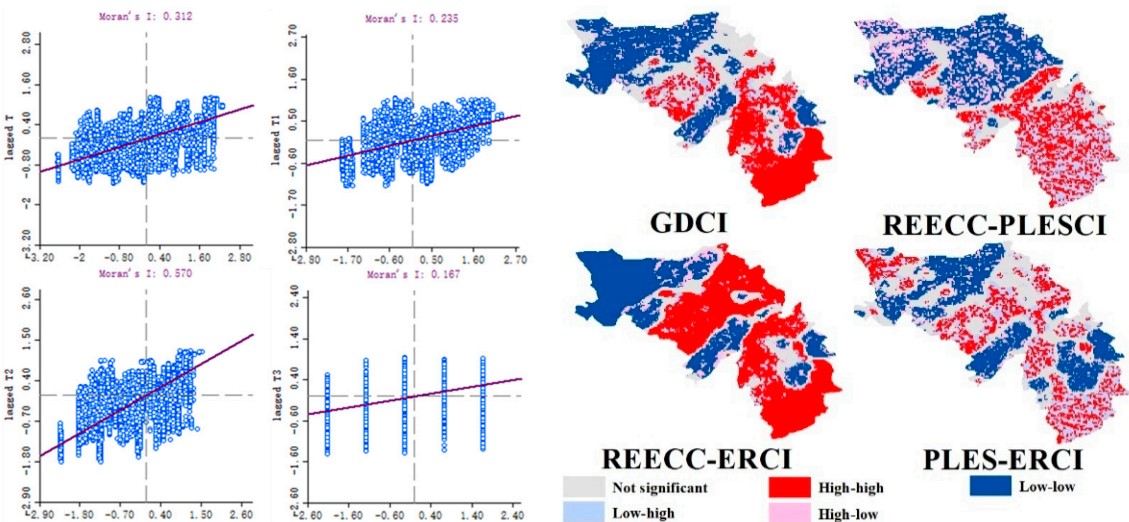

**Figure 7.** The global Moran's index and local Moran's index of green development composite index.

From a spatial point of view (Figure 8), the coupling degree of green development includes five types: extremely low coupling stage, low coupling stage, middle coupling stage, high coupling stage, and extremely high coupling stage. The degree of coupling coordination includes moderate balance, balance, basic coordination, and moderate coordination. During the study period, the coupling degree and coupling coordination degree of green development showed the spatial distribution characteristics of "large-scale high-value distribution area–multipoint median distribution area–small-scale low-value continuous area". More than 80% of the regions have a high coupling and coupling coordination state. This indicates that REECC, PLES, and ER have mutual driving effects.

In addition, the coupling relationship and interaction intensity between REECC, PLES, and ER have significant spatial differences. The areas with a high coupling coordination degree are mainly scattered in north of the Dongliao River Basin, specifically in Lishu County and Gongzhuling City. The counties with a low coupling coordination degree are mainly concentrated in the Siping and Liaoyuan urban areas. The coupling degree presents a spatial pattern that gradually increases from the urban area to the outside.

The degree of coupling and coordination coupling between REECC and ER is the highest, and the spatial distribution gradually decreases from northwest to southeast. However, the coupling coordination degree between REECC and PLES shows the opposite trend, with the northwest being more coupled and the southeast being less coupled. The coupling coordination degree between PLES and ER is dense and high in Yitong County, Siping City, and Shuangliao City.

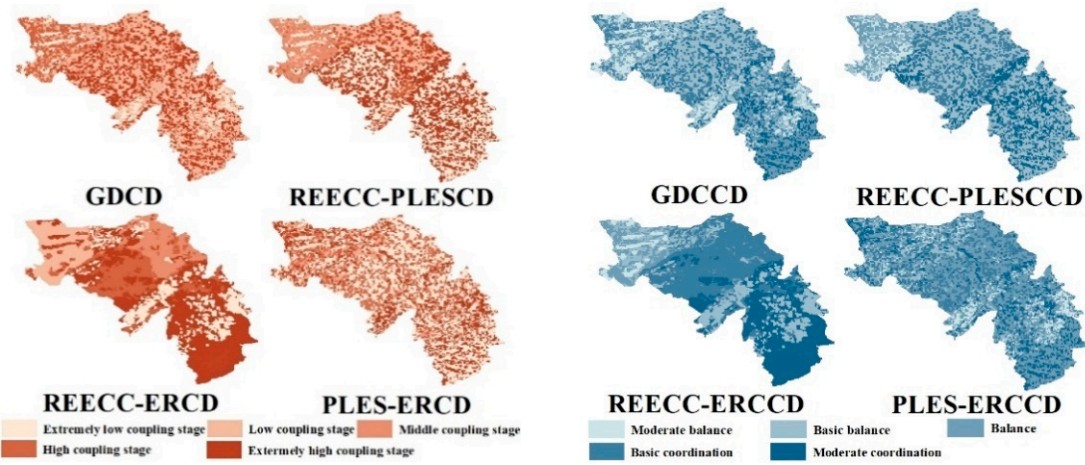

**Figure 8.** The spatial distribution of coupling degree and coupling coordination degree.

The spatial form of LISA (Figure 9) shows that the global Moran index is positive, but the coupling degree and coupling coordination degree among different systems show some differences in the Dongliao River Basin. The HH gathering area is mainly flaky around Siping City and Liaoyuan City, while the LL gathering area is mainly located in Shuangliao City, west of Gongzhuling City, and the center of Siping City and Liaoyuan City, with a high distribution correlation. It is worth noting that the spatial correlation of the comprehensive index of green development is enhanced, which indicates that green development is strongly influenced by human interference.

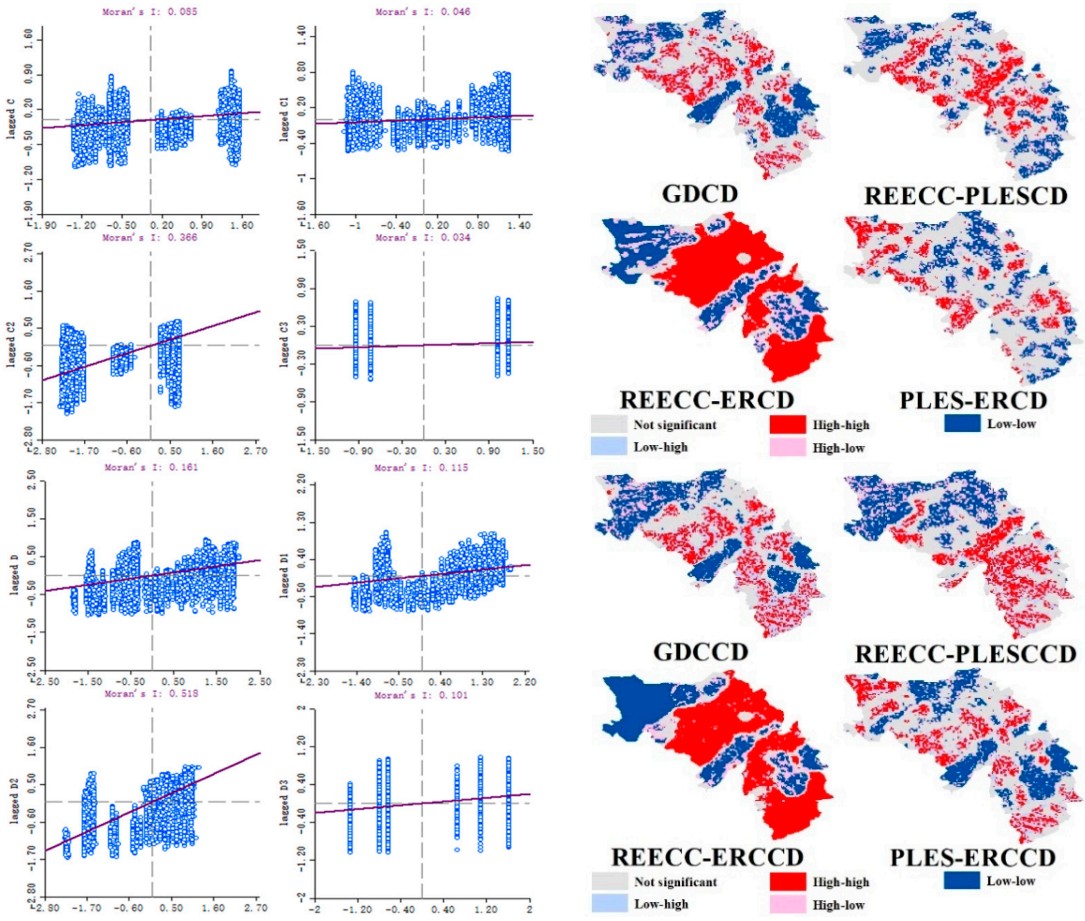

**Figure 9.** The global Moran's index and local Moran's index of coupling degree and coupling coordination degree.

## 5. Conclusions

Based on the new perspective of regional green development, the evaluation system of resource, environment, and ecological carrying capacity and the environment is constructed, which includes 14 indicators. For the calculation of the weight of the indicator, we used the method of combining principal component analysis and a geodetector model. The comprehensive green development index in DRB was calculated by this evaluation system. According to the formula of coupling degree and coupling coordination degree, the coupling coordination relationship of green development in 2018 was calculated. The conclusions of this paper are as follows:

(1) The status of sustainable development and resource, environment, and ecological carrying capacity are the dominant factors affecting the GD level of each district and county in the Dongliao River Basin. The "soil and water conservation", "water conservation", "land desertification", and "soil and water loss" are the indicators with the largest weight in the evaluation index system of resource, environment, and ecological carrying capacity.

(2) There are significant spatial differences in REECC in the Dongliao River Basin. The carrying capacity index is gradually decreasing from the southeast to the northwest.

(3) In 2018, the overall level of green development in the DRB has obvious spatial dependence. However, there are differences in spatial distribution: the level of green development gradually increases outward with the city as the center and presents a trend of gradually decreasing from southeast to northwest.

(4) The spatial distribution of the coupling degree and coupling coordination degree is roughly the same, but there is a cluster distribution. The leading factors of GD in the Dongliao River Basin are ecological redline and production–living–ecological spaces. The empirical results show that the evaluation model and method can be used to simulate and evaluate the level of regional green development in China due to its scientific basis and relevance, indicating that the study results are time-sensitive.

Therefore, this study proposes strategies to promote and improve regional green development. The strategy is guided by the construction of ecological civilization, and aims to promote the green development of the basin in the future from the perspective of coordinated development of resources, environment, ecology, and social economy:

(1) The government agencies should pay attention to increasing investment in scientific research, ensuring sufficient research funds, and promoting the transformation of scientific and technological achievements into productivity. Environmental protection technologies, such as soil and water conservation used to increase the potential value of regional green development, should be focused on for strengthening by managers.

(2) Strive to create a climate of environmental protection, develop recyclable and other environmental protection products, and strengthen the construction of ecological civilization. In addition, an information network covering resources, environment, ecology, land use, online utilization of resources, environmental quality bottom lines, and ecological redlines should be established to effectively and efficiently disseminate regional environmental protection information and promote the building of a greener society.

(3) The government should vigorously promote reform policies, including regulating the market, improving laws and regulations, establishing a green industrial chain, and advocating green consumption, etc., to provide a good development space for green enterprises and organizations to invest in environmental protection, diversify environmental investment, and increase participation in green development.

**Author Contributions:** Conceptualization, X.L.; Funding acquisition, J.Z.; Investigation, W.D.; Methodology, A.W.; Project administration, J.Z.; Resources, Z.T.; Software, X.L.; Supervision, Z.Y.; Validation, W.D.; Writing—original draft, A.W.; Writing—review & editing, Z.T. All authors have read and agreed to the published version of the manuscript.

**Funding:** This research was funded by the Major Scientific and Technology Program of Jilin Province, (China) (20200503002SF); the Science and Technology Development Planning of Jilin Province, (China) (20190303081SF).

**Acknowledgments:** The authors would like to thank the Data Center for Resources and Environmental Sciences, Chinese Academy of Sciences (RESDS, http://www.resdc.cn) (accessed on 26 March 2021) for providing the land use dataset.

**Conflicts of Interest:** The authors declare no conflict of interest.

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
