# Peer review of "Comprehensive Evaluation of Green Development in Dongliao River Basin from the Integration System of “Multi-Dimensions”"

_sustainability, doi:10.3390/su13094785_

Round 1
Reviewer 1 Report
I found the reviewed paper study as interesting and up-to-date one. There are some issues arise to the research which require explanation and/or farther research:
- the sustainable development and green development (GD) concepts have not been distinguished in detail clearly
- because of (1) the classification of the indicators applied by the Authors is rather controversial
- the dimensions of resources, environment and ecology are not clearly defined. (see e.g. verses 66, 82-83). The resources are natural environmental assets as well as ecological cycles
- what does it mean “in 2018 the overall level of GD was relatively high”? (v.413-414). Relatively comparing to what reference data?
Author Response
Please see the attachment.
Dear reviewer:
Thanks very much for taking your time to review this manuscript. We appreciate all your generous comments and suggestions! Please find my itemized responses in below and my revisions in the re-submitted files.
1.Response to comment: the sustainable development and green development (GD) concepts have not been distinguished in detail clearly.
Response: We added concepts and research progress on sustainable development and green development in line 41-42 and line 45-47 of the manuscript to make the difference more clearly.
- Response to comment: the classification of the indicators applied by the Authors is rather controversial.
Response: We added policy criteria for selecting evaluation indicators of resource, environmental and ecological carrying capacity in line 331-334 of the manuscript in order to make this process more scientific and objective.
- Response to comment: the dimensions of resources, environment and ecology are not clearly defined.
Response: After referring to your opinions, we defined the dimensions of resources, environment, and ecosystem based on the system dynamics model and analyzed their differences and connections. The analysis results are added to line 210-215.
- Response to comment: The conclusion "in 2018 the overall level of GD was relatively high" lacks reference data.
Response: We are sorry for our incorrect writing in conclusions. After reading a lot of literature, we found that there was no effective support, so we revised this conclusion to "In 2018, the overall level of green development in the DRB has obvious spatial dependence". The revised results are in line 450-451 of the manuscript.
In all, I found the reviewer’s comments are quite helpful, and I revised my paper point-by-point. Thank you and the review again for your help!
Sincerely yours,
Aoyang Wang

Reviewer 2 Report
The article is well organized and developed, both in terms of content, methodology and results. However, there are some aspects that could be improved.
In the abstract, the phrase "decreasing trend from southeast to northwest" is repeated in lines 21-22 and 23-24. Although it refers to different topics, it would be convenient to change some words so that it does not seem repetitive in a very short heading. Moreover, when citing the cardinal points, it is more correct to express it as "from northwest to southeast" (as the authors do elsewhere in the article).
In the abstract, the phrase "decreasing trend from southeast to northwest" is repeated in lines 21-22 and 23-24. Although it refers to different topics, it would be convenient to change some words so that it does not seem repetitive in a very short heading. Moreover, when citing the cardinal points, it is more correct to express it as "from northwest to southeast" (as the authors do elsewhere in the article).
Bibliographic reference 10 does not appear in the text, only in the bibliography, and the text skips from 9 to 11.
In epigraph 2.1, it is indicated that the study area is highly populated and with serious environmental problems: water pollution, erosion, decrease of forests and wetlands, backward environmental management models, economic growth restricted by irrational natural and human activities... top priority of ecological restoration reconstruction even of the whole country. However, in both the discussion and conclusion it is stated that "More than 80 % of the regions have a high coupling and coupling coordination state" (line 370) and in the conclussion (3) "In 2018, the overall level of Green Development in the Dongliao River Basin isrelatively high".
"Resources, environment and ecosystem" are mentioned on several occasions. It would be useful to clarify these concepts, since they are very overlapping, especially the last two. Perhaps a footnote could be included to avoid confusion for other researchers.
The diagrams are understandable, but they are quite dense and contain a lot of information, although the processes are also complex. It would be convenient to specify whether they are their own, adapted or from other authors.
In 2.3.2 the principal component analysis is mentioned, but the statistical results are not shown, except in a generic way. The variables used and the conformation or structure of the two or three principal components should be indicated. Likewise, the statistical results of the Moran indices are not detailed, although the LISA indices are mapped. It would be interesting to expose these statistical results, in order to complete the text and demonstrate the results in a more reliable way for a better replication in other places and by other researchers.
In lines 272-274 it is indicated that there are three subsystems, but four are mentioned: resource carrying capacity, environmental carrying capacity, ecological carrying capacity, and socio-economic carrying capacity", It is repeated in lines 385-386.
All the bibliographical references, except for those referring to statistical techniques, are by Chinese authors. It is logical, but for the development of the theoretical and conceptual headings, other external authors should have been consulted, who could have enriched it. There are some very good works, some of them published in this review.
Author Response
Please see the attachment.
Dear reviewer:
Thanks very much for taking your time to review this manuscript. We appreciate all your generous comments and suggestions! Please find my itemized responses in below and my revisions in the re-submitted files.
1.Response to comment: the phrase "decreasing trend from southeast to northwest" is repeated and the incorrect expression.
Response: Thank you for your detailed comments and explanations. We modified and corrected this phrase, see line 22-24.
- Response to comment: Bibliographic reference 10 does not appear in the text, only in the bibliography.
Response: After re-reading the reference 10, I found that it was indeed as you suggested, so I deleted this reference and added a new reference on line 49 of the manuscript.
- Response to comment: The conclusion "More than 80 % of the regions have a high coupling and coupling coordination state" and "In 2018, the overall level of Green Development in the Dongliao River Basin is relatively high" are inconsistent with the research background.
Response: We attach great importance to this comment. After reading a lot of literature and reports, we revised the research background: "In the period of rapid socio-economic development in the 20th century, the area has experienced serious water pollution, reduced resource and environmental capacity , shrinkage of wetlands and forests, soil erosion, unreasonable resource allocation, and backward ecological environment management models which in turn have restricted the economic growth of the region due to natural and human irrational activities. As a key river basin in Jilin Province and even the whole country, the Dongliao River Basin has been regarded as a demonstration area for ecological restoration since 2007. At present, the pollution situation in the area has been preliminarily controlled and scientifically treated." See line 163-171. Due to the lack of comparative data, we also revised the conclusion to "In 2018, the overall level of green development in the DRB has obvious spatial dependence". See line 450-451.
- Response to comment: Resources, environment and ecology are not clearly defined.
Response: After referring to your opinions, we defined the dimensions of resources, environment, and ecosystem based on the system dynamics model and analyzed their differences and connections. The analysis results are added to line 210-215.
- Response to comment: The sources of the diagrams are unknown.
Response: I'm sorry for not clarifying the sources of the diagrams in the manuscript. They are all original, and I illustrate this point in line 214, line 337 and line 380-381 in the manuscript. Thanks again for your comments!
- Response to comment: Principal component analysis and Moran’s index statistics are not shown.
Response: Thank you for pointing out the shortcomings of our research. After careful verification and correction, we added a text description about principal component analysis in line 332-333 and added the principal component analysis results in line 335, which is Table 2. In order to show the statistics of the Moran’s index, we added a description of the local Moran’s index in line 381 and added a local Moran’s index scatter plot in Figure 7 and Figure 9.
- Response to comment: Wrong writing of the number of subsystems.
Response: Thank you so much for your careful check. We have changed the wrong three subsystems to four in line 302
- Response to comment: Theories and concepts lack the research of other external authors
Response: We think your comments are very helpful to us, so we have cited many other external authors' reference when discussing the concept of sustainable development (line 42) and the development of the concept of green development (line 46).
In all, I found the reviewer’s comments are quite helpful, and I revised my paper point-by-point. Thank you and the review again for your help!
Sincerely yours,
Aoyang Wang
